# Hai||om children mistrust, but do not deceive, peers with opposing self-interests

Roman Stengelin[1,2]*, Robert Hepach[1,3], Daniel B. M. Haun[1,2,4]

**1** Leipzig Research Center for Early Child Development, Leipzig University, Leipzig, Germany, **2** Department of Early Child Development and Culture, Leipzig University, Leipzig, Germany, **3** Department of Research Methods in Early Child Development, Leipzig University, Leipzig, Germany, **4** Department for Comparative Cultural Psychology, Max-Planck-Institute for Evolutionary Anthropology, Leipzig, Germany

\* roman.stengelin@uni-leipzig.de

## Abstract

During their preschool years, children from urban, Western populations increasingly use deception and mistrust to regulate social interactions with others who have opposing interests. The ontogeny of these behaviors in rural, non-Western populations remains understudied. This study assessed deception and mistrust within peer interactions among 4- to 8-year-old Hai||om children from rural Namibia ($N = 64$). Participants engaged in a dyadic game in which their self-interests were either aligned (cooperation condition) or opposed (competition condition) to those of their coplayers. Similar to previous evidence taken from Western participants, children mistrusted their coplayers during competition, but not during cooperation. Rates of actual deception were low in both conditions, which contrasts previous findings among Western populations. On an individual level, those children who deceived were also more likely to mistrust their peers. These results reveal novel insights on the ontogenetic primacy of mistrust over deception in young children's peer interactions in a rural, non-Western community.

## Introduction

The propensity to utilize information provided by others is an essential characteristic of our species [1,2]. By relying on others' testimony, we can circumvent costly trial-and-error learning. However, not every informant is helpful and benevolent. Individuals need to understand the circumstances under which they can rely on others' testimony and, accordingly, learn whom to trust and whom not to trust.

One crucial step in the ontogeny of selective trust is that children need to learn that an informant's credibility relies on situational incentives. Around age 5, children start to mistrust informants with self-interests that are opposed to their own–such as coplayers in a competitive game–and they hone this selective mistrust increasingly throughout their preschool years [3–5]. This developmental of children's selective mistrust is accompanied by an increase in their use of strategic deception. While children tell their first lies around age 3, their abilities to deceive others continue to become more sophisticated throughout middle childhood [6,7]. Theoretically, the ontogeny of selective trust based on incentives and deception should go hand in hand [4,8], since mistrusting others is a consequence of anticipating that the

**Data Availability Statement:** Data is given in the supporting information.

**Funding:** The work was funded by internal budgets provided by the Department of Early Child Development and Culture at Leipzig University. No

other parties than the authors had a role in study design, data collection and analysis, decision to publish, or preparation of the manuscript.

**Competing interests:** The authors have declared that no competing interests exist.

informant may intend to deceive. However, empirical support for this assumption is currently sparse [4].

Previous studies have mostly assessed children's selective trust and deception in the context of *adult-child interactions* (or between a child and a puppet played by an adult experimenter). However, children's social learning is not limited to learning from adult testimony. Peers also constitute an essential source of information throughout childhood [2,9], even though children remain biased to trust adult testimony over that of peers under most circumstances [10]. Assessing mistrust only during adult-child interactions may thus systematically underestimate children's capacities. A similar issue applies to children's deception of adult coplayers. Children may have some reservations to deceive adults due to power imbalances: Their capacities to demonstrate mistrust and deception in interactions with adults may thus be impeded by a bias to be honest and trust adults out of social motives such as politeness or obedience.

Previous work furthermore typically assessed children's mistrust and deception during interactions with an *unfamiliar* adult. The potential effects of this procedural aspect may be twofold. On the one hand, doing so may lower reputational and moral concerns since children do not anticipate future interactions with the coplayer [11]. This lowered emphasis on reputation may lead children to both mistrust and deceive at higher rates than what one would expect in situations comprising familiar informants. On the other hand, it might be more difficult to mistrust or deceive unknown individuals, since shared experiences with friends and peers might help to anticipate their behaviors more efficiently. In any case, systematic assessments of young children's mistrust and deception toward familiar interactants are needed to understand the early ontogeny of both strategies under more ecologically valid conditions.

Finally, prior research has almost exclusively focused on participants from *urban*, *Western populations* [12,13]. The majority of studies on intent-based selective trust have been conducted among urban participants from the U.S. [3,4] or Western Europe [5,14]. This sampling bias renders generalizations outside such cultural contexts difficult, if not invalid [13]. To fully understand the ontogeny of trust and deception in humans, examinations of the two phenomena in more rural, subsistence-based contexts is highly needed. Contemporary hunter-gatherer societies are of particular interest in this endeavor, given that this form of subsistence, including its socio-cultural covariates, is the closest contemporary approximation of the circumstances under which human psychology evolved. The Hai||om of northern Namibia are recent former hunter-gatherers and may thus be highly informative with regards to the ontogeny of mistrust and deception within peer interactions. Today, the Hai||om increasingly practice a mixed economy in which hunting and gathering is combined with agriculture, the selling of handmade crafts, and occasional wage labor [15]. However, those living in more rural areas are still describing their cultural identity as that of hunter-gatherers, and their traditional values and socialization goals are still practiced and considered valuable. One foundational norm that the Hai||om and other hunter-gatherer societies typically emphasize is that of sharing on demand [15,16,2]. Accordingly, individuals are obliged to share resources among members of the community upon demand from an early age onwards [16]. This norm may arguably shape the ontogeny of deception and mistrust such that more cooperative, rather than competitive, communication strategies may be promoted when facing limited resources. That is, if obtained resources are shared among peers regardless of individual merit or success [16], incentives for deception (and, as a consequence, mistrust) should be lowered markedly as compared to societies emphasizing merit and individual attainments. Accordingly, such egalitarian sharing norms may create cooperative incentives under conditions that would be perceived as competitive among Western populations.

The current study examined mistrust and deception among young Hai||om children. Children played either a cooperative or competitive version of a sender-receiver game [4]. During

cooperative trials, the self-interests of both coplayers were aligned (winning or losing together). During competitive trials, coplayers had opposing interests, as the success of one player came along with the loss of the second player (and vice versa). All children played two roles throughout of the game: the *sender* was shown the secret location of a reward and could then give a hint to the *receiver*, who could choose where to look for the reward. Children took turns in playing the two roles without receiving feedback on whether their former strategy was successful, allowing us to assess both deception (hints of the *sender*) and mistrust (the choices of the *receiver*) for each child.

Based on previous evidence from Western participants, we preregistered the following hypotheses on the Open Science Framework (OSF; https://osf.io/b8gja/?view_only= b8f0f8c9499b4e6c97a3a541b2fe4b68): First, we expected that *senders* would deceive *receivers* more often when playing under competitive incentives as compared to playing cooperatively (Hypothesis 1a). We predicted this effect to increase with age. Further, we assumed *receivers* to mistrust *senders* more often when playing the game under incentives implying competition as compared to cooperation (Hypothesis 2a). This tendency should increase with age. Regardless of whether children would show an effect of condition or not, we predicted children's mistrust and deception to be positively (Hypothesis 3).

If, in contrast, deception and mistrust would be of little relevance among the Hai||om due to their emphasis on sharing norms, one would expect *senders* to indicate the correct location of the reward regardless of condition (Hypothesis 1b) in order to assist their coplayer in finding the reward (see also [17]). If so, one would also expect *receivers* to trust their coplayers (Hypothesis 2b) regardless of game incentives out of anticipation of their coplayers' honest advice.

## Methods

### Participants

We tested 32 dyads of children ($N$ = 64 children, $M_{Age}$ = 6.94, 50% female, age range: 4 to 8 years). Children within dyads were matched for age ($M_{Diff}$ (SD) = 0.47 years (0.44)) and sex. Siblings or children living in one household were not tested together in the same dyad. Siblings or children living in one household were not tested together in the same dyad. All participants were Hai||om children from Ondera, a rural village with around 800 inhabitants in northern Namibia. While some people at Ondera find employment at the local vegetable farm, others sell handmade crafts at Etosha Park and cultivate plants in their gardens. Gathering bushfood is practiced commonly by the Hai||om at Ondera. Hunting is forbidden by law. In line with Hai||om traditional parenting practices, children are given high levels of autonomy from early on [15]. As is common in hunter-gatherer societies, children's daily activities are typically structured within mixed-age peer groups, which is why peer interactions can be considered essential for social learning [18].

At Ondera, children can visit a local primary school from age seven onwards. Younger children can attend a kindergarten run by local Hai||om caregivers. All children who participated in the present study were enrolled in either the local school or kindergarten. Participants were well-known to their coplayers to ensure that the dyadic study context would resemble the social interactions children engage in their daily lives at Ondera.

Legal requirements of the Republic of Namibia were strictly adhered to. The research was approved by the Ministry of Home Affairs and Immigration in Namibia and the Working Group of Indigenous Minorities in Southern Africa (WIMSA). Further, the Ethical Committee at the Medical Faculty, Leipzig University approved the research (title of the approval: "Investigating the non-pathological development of social behavior and competences in children and

adults by using behavioral, peripheral physiological, and psychometric methods"; reference number 169/17-ek). Parental and school-principal's informed consent was obtained prior to the study.

## Materials & design

Dyads of children played a game in which they were asked to find candy (skittles) hidden in one of two locations. In a between-subjects paradigm, dyads were randomly assigned to one of two experimental conditions that differed regarding both players' incentives: In the cooperation condition, both children played the game with aligned self-interests. In the competitive condition, self-interests were conflicting. After training, children took turns in giving a hint about the location of a reward to their coplayer (*sender*), and in deciding whether or not to follow these hints when looking for the candy (*receiver*). Each child played both roles twice during test phase. The location of rewards was counterbalanced across trials and dyads.

Children sat on grey cushions facing one another. Candies were hidden in blue plastic balls (diameter: 3cm), placed on white plates on the floor between the participants (see Fig 1). These balls either contained candy (Skittles as rewards) or small stones of similar size and weight (no rewards). After children made their choices, balls were stored in blue plastic buckets. At test, hints could be provided by placing a wooden stick on one of the plates. Children collected their candies in paper bags.

Study instructions were initially composed in English by the first author and independently translated into Hai||om by two local research assistants. Translations were then compared and discussed between both translators and the first author to resolve eventual disagreements.

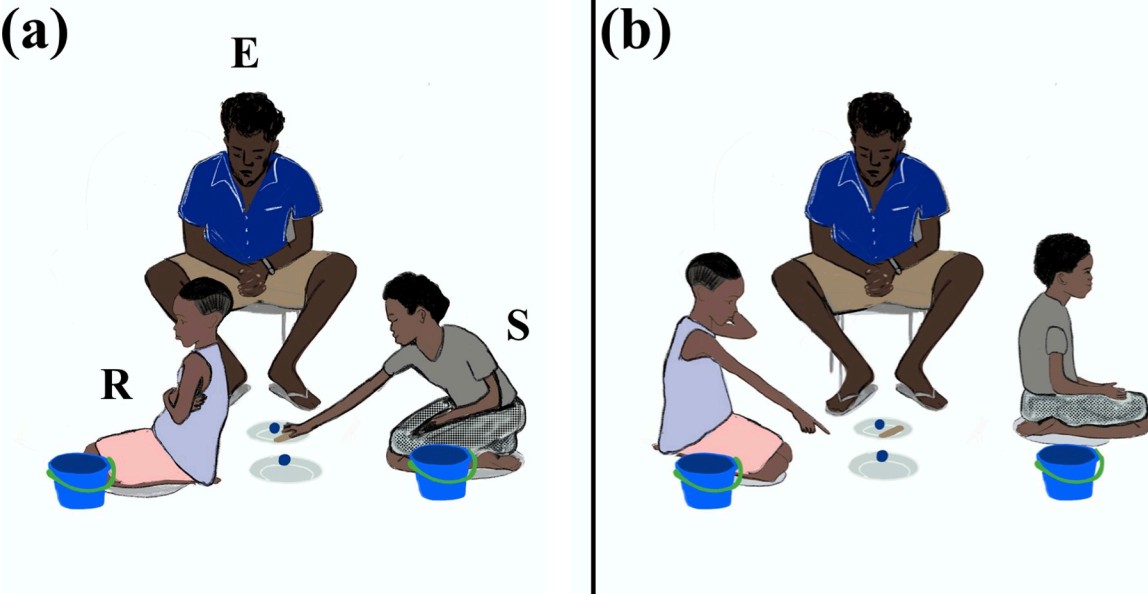

**Fig 1.** Set-up during test trials (competition condition is illustrated in which balls are stored in buckets separately for each child); E = Experimenter; S = Sender; R = Receiver; (a) S places wooden stick on the plate next to E (deception or truth assessed based on the correct location of the reward); (b) R indicates her choice while S turns away (mistrust or trust assessed depending on the hint given by S, here: R mistrusts S).

## Procedure

Testing took place in a room at a local school building, led by a local male experimenter (E). Before children entered the study room, they were asked whether they would like to participate and informed that they could stop participating at any time throughout the procedure.

**Instruction phase.** E guided both participants to the testing room and introduced the rules of the game. In the cooperation condition, he explained that children could win some candy in the game by finding the candy that was hidden in one of two plastic balls. E presented two balls and opened them in front of the children. One ball contained two candies, whereas the other ball contained a stone. E emphasized the same look and sound of the two balls and explained that it was impossible to know which of the balls contained the candy unless one would open them. Next, E placed two other balls on the plates between the coplayers. He explained that children could take turns in guessing where the candy was hidden. In each trial, guessers would signal their choice by pointing to one of the two plates, before E would place this ball into the shared bucket of both coplayers. The remaining ball would then be put into another bucket next to E. The coplayer who would not guess in this trial would turn her back during the scene such that guessers would make their choice in private and without being observed by their coplayers. If children attempted to peak to see their partner's choice during this phase, they were reminded by E to turn away to ensure children's privacy in making their decisions. After each trial, children would get immediate feedback on whether the choice of the guesser was successful or not. If children indicated the correct ball, E gave one candy to each participant. If guessers would choose the wrong location, no one received a reward. In the competitive condition, instructions were similar except that the correct ball contained one candy, and each child had an own bucket to collect the balls. E explained that guessers would receive the candy if they found the reward. If not, their coplayers would get the candy instead.

To ensure children's comprehension of the game's incentives, E asked each participant who would receive the candy if they found the reward (correct answer in cooperation condition: Both of them; correct answer in competition condition: Child herself). Second, he asked who would receive the reward if they would make an incorrect guess (correct answer in cooperation condition: Neither of them; correct answer in competition condition: Coplayer). If one of the participants gave an incorrect answer, E explained the rules of the game once more, before repeating the comprehension questions. Only dyads in which both children answered the comprehension questions correctly were included in the study. Two dyads did not solve these questions and were thus not examined any further.

**Guessing game I–training with immediate feedback.** After the instruction phase, children started to play the game. Child one was first to choose between the two balls by guessing. In the meantime, child two turned her back toward the scene. After child one had made her choice, E took the ball and put it in the bucket of child one (competition condition) or the shared bucket of both coplayers (cooperation condition). Then, he took the remaining ball and put it into a second bucket (coplayer's bucket in competition condition, remaining bucket in cooperation condition). Child two was asked to turn back toward the scene so that both children could observe E opening both balls. Children could store their rewards in their paper bags. Each child played the role of the guesser twice in this training phase, resulting in 4 training trials.

**Guessing game II–training with delayed feedback.** E asked the comprehension questions again to remind children of the game's incentives. In deviation from prior trials, E introduced a new rule to the game. Instead of receiving immediate feedback on their choices, children would collect the balls they (or their coplayer) chose in the respective buckets, and rewards would only be extracted after four consequent trials. The purpose of introducing this rule was to make sure that children would not be able to identify whether their choice in a

given trial was correct or not, which was central for the subsequent test phase. Children did not get any hints regarding the true location of the reward but had to guess instead. The balls were rigged such that every child would receive candy for two out of the four training trials.

**Sender/receiver game–test.**   E introduced a new rule to the game. From now on, one child could see the correct location of the reward. This child (henceforth *sender*) could provide a hint for the coplayer (henceforth *receiver*) who could then utilize this hint to guess the location as in previous trials. Senders were given to opportunity to give a hint by placing a wooden stick on one of the plates on which the balls were presented. E asked the comprehension questions again to ensure that both coplayers understood the game's incentives. Before each trial, receivers turned around so that E could show the content of each ball to senders secretly. E then closed the balls and asked senders to place the wooden stick on one of the two plates (see Fig 1A). Next, receivers could join the game while the sender was asked to turn around. Receivers made their decisions in private (see Fig 1B) and E stored the balls in the buckets according to receivers' choices. Again, feedback was given after four trials to make sure that children did not know whether their coplayer had mistrusted or deceived them in previous rounds. Children received their rewards following four test trials.

## Data coding

Children's choices were coded from video by the first author. For reliability coding, the full sample was coded by a second coder blind to conditions and hypotheses. Interrater agreement regarding both children's deception ($\kappa = 0.98$) and their mistrust ($\kappa = 0.98$) was excellent.

## Statistical models and preliminary analyses

Data analyses were conducted in *R* [19]. To investigate Hypotheses 1 and 2, we ran a generalized linear mixed model (GLMM) with binomial error structure using the package *lme4* [20]. We estimated the statistical significance of each predictor using a likelihood ratio tests comparing a full model to a reduced model without the respective predictor. Data and code for all analyses are available in the *supplemental materials.*

**Deception (H1).**   We investigated whether participants deceived their coplayers depending on two predictors: Condition (cooperation vs. competition) and age (continuous variable). Sex and trial number were included as control variables. We further included ID as a random intercept to account for within-subject variance. Dyad was not included to avoid singularity issues. The number of rewards children received during Guessing Game I was not included to the model because of convergence issues. In a separate analysis, this variable was not linked to children's deception ($\chi^2$ (1) = 0.354, $p$ = .552).

**Mistrust (H2).**   We investigated whether children's mistrust depended on the same predictors: Condition (cooperation vs. competition) and age (continuous variable). We included the same intercepts and controlled for the same variables as in the deception model. The number of rewards children received during Guessing Game I did not predict their subsequent mistrust ($\chi^2$ (1) = 0.327, $p$ = .568).

**Deception and mistrust (H3).**   To analyze whether deception and mistrust would relate to one another on an individual level, we calculated the correlation between both behaviors.

## Results

### Deception (H1)

The interaction between age and condition did not reach statistical significance ($\chi^2$ (1) = 0.05, $p$ = .827), which is why we describe the final model with main effects only. Children were not

significantly more likely to deceive their coplayer in the competition condition as compared to the cooperation condition ($\chi^2$ (1) = 1.08, $p$ = .299, see Fig 2A). Age did not affect children's deception ($\chi^2$ (1) = 1.39, $p$ = .238).

### Mistrust (H2)

The interaction between condition and age did not reach statistical significance ($\chi^2$ (1) = 0.17, $p$ = .682) and was thus dropped from the model. Children playing the game with competitive incentives were more inclined to mistrust their coplayer's hints than those playing cooperatively ($\chi^2$ (1) = 7.18, $p$ = .007, see Fig 2B). Age did not significantly affect children's deception ($\chi^2$ (1) = 0.47, $p$ = .492).

### Deception and mistrust (H3)

Children who deceived their coplayer were also more likely to show mistrust ($r_\varphi$ = .19; $p$ = .036).

## Discussion

In the current study, we found that children from a recent former hunter-gatherer community selectively mistrusted advice provided by their peers in a competitive context. However, they deceived only rarely and regardless of their coplayer's incentives. Children's deception and mistrust were linked on an individual level.

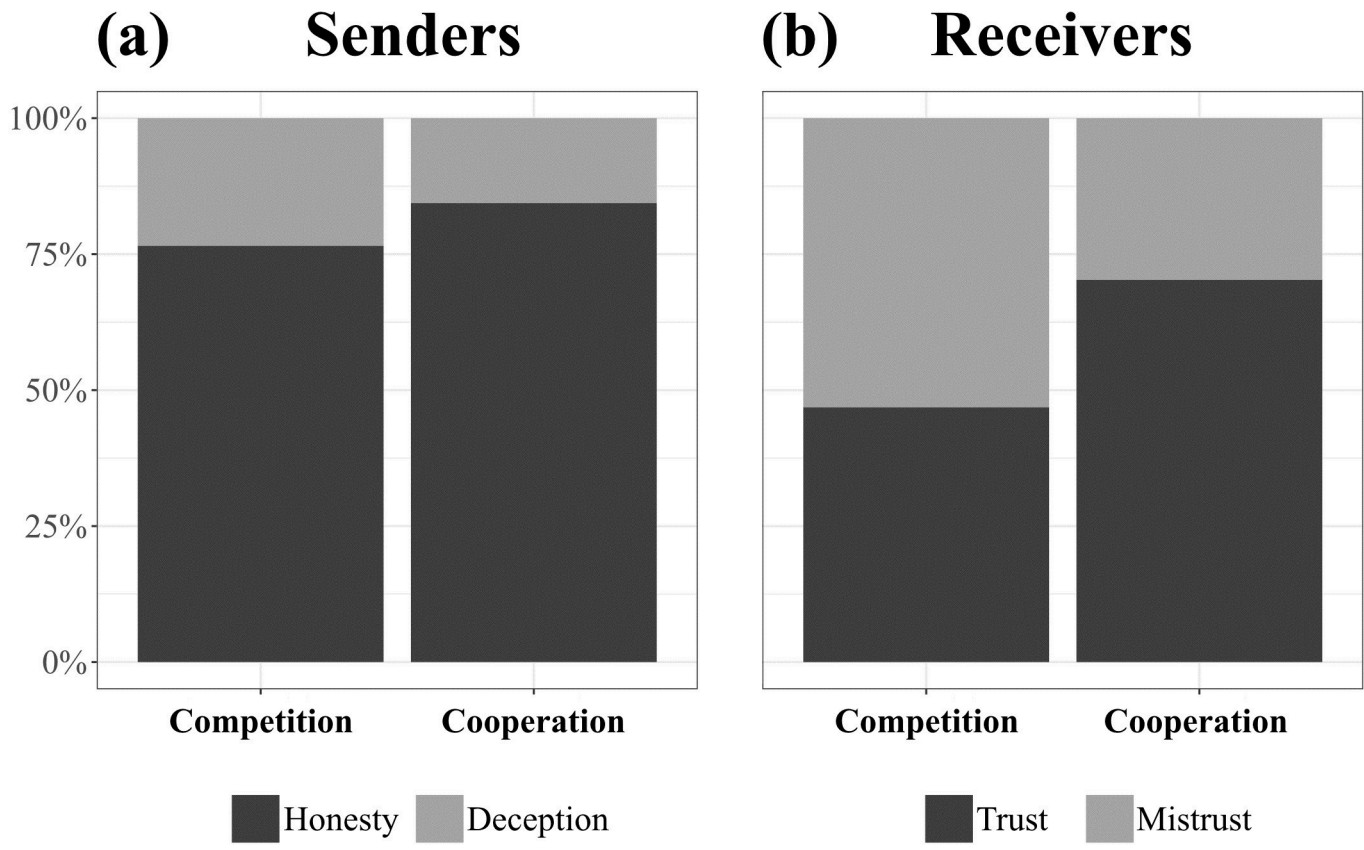

**Fig 2.** Children's behaviors throughout the study; (a) honesty vs. deception as senders; (b) trust vs. mistrust as receivers.

These findings differ from previous studies assessing the early ontogeny of mistrust and deception based on procedures incentivizing competition. In previous studies, preschool children from urban Western populations were found to both mistrust and deceive their coplayers at similarly high rates [3,4]. While children in the current study mistrusted their competitive coplayers at rates comparable to previous studies, they used deception less frequently and irrespective of whether game incentives favored competition or cooperation. Moreover, neither mistrust nor deception became more frequent with age, which also stands in contrast to previous work [3–5]. Procedural details may account for these diverging results: While previous studies introduced either unfamiliar adults [4] or puppets played by adults [3,5] as coplayers, children in the current study interacted with a familiar peer. It is plausible that senders provided receivers with honest cues when playing with a competitive incentive because of reputational concerns.

Moreover, this study was the first to assess children's mistrust and deception in a rural, non-Western population. In combination, these factors may have led senders to give honest hints out of reputational concerns. It is plausible that deceiving peers, and friends in particular, is seen as a more severe transgression of social conventions than doing so when interacting with strangers [11]. Such concerns may be of higher relevance among rural as compared to urbanized communities, given the importance of long-lasting social bonds. Furthermore, in the current study, receivers could make their decisions whether to trust or mistrust their coplayers in private, given that senders turned away from the scene. This procedural detail may thus have lowered reputational concerns for children when playing as receivers.

The societal emphasis on sharing on demand among the Hai||om may also be essential to understand why senders in this study did not deceive receivers at higher rates. Accordingly, children may have hinted at the correct location of the reward because they did not adopt a competitive stance toward their coplayer in the game. Children may have expected that resources obtained throughout the game would be shared afterward regardless of whether they or their coplayer would obtain the reward. This expectation may have led them to give honest hints to avoid frictions. However, this conclusion fails to explain children's mistrust in the role of receivers. If children had a cooperative stance in general and regardless of condition, one would expect them to trust their coplayer's hints in either condition.

Alternatively, children may have provided truthful information to their coplayers because they lacked the social-cognitive skills needed for deception, such as false belief-reasoning [4,21]. However, children's selective mistrust as receivers suggests that they understood that the game's incentives might lead their coplayer to deceive them. As such, the low rates of deception in the current study are unlikely due to a lack of skills in false belief-reasoning. To our knowledge, there is no documentation of how false belief-reasoning emerges develops among Hai||om children. The one study that has assessed the ontogeny of this skill among hunter-gatherer children has documented sophisticated skills in false belief-reasoning among Aka children [22], which is in line with other evidence suggesting that such reasoning is common among young children from rural, non-Western societies ([23,24], but see [25]). Future studies will need to assess if the links between false belief-reasoning and deception (and mistrust) observed among Western children [21] recur among non-Western societies. Based on the current findings, it is unlikely that a lack of skills in false belief-reasoning is the reason for the low rates of deception in the competition condition.

A more plausible explanation may be that senders used more sophisticated deception strategies because they anticipated that their coplayer would mistrust their hints. If so, indicating the correct location would be an efficient and deceptive strategy for senders. Such considerations, which rely on recursive false belief-reasoning skills, have been observed to increase with age in Western societies [4]. At first glance, this explanation may also be in line with the fact that we did not find any effect of age in our data. If younger children did not yet engage in

deception at all, older children may have chosen the same behavior because they anticipated that their coplayer would likely mistrust their advice. If so, however, children's deception during competition should have first increased with age before converging to chance levels at older ages. The paradigm introduced in this study cannot identify the cases in which a sender would give an honest hint from those in which she would anticipate her coplayer's mistrust via recursive false belief-reasoning. Novel paradigms are needed to disentangle these strategies in order to get a thorough understanding of the early ontogeny of deception and mistrust.

The absence of any age-related change in children's mistrust and deception in the current study may be best explained regarding the ontogeny of reputational strategies in young children. At age 5, children from Western populations already care about their reputation such that they cheat less and act more prosocially in the presence of others ([21], see also [22]). It is plausible that this sets the standard for mistrust and deception when interacting with familiar peers, such that ontogenetic differences in the use of either strategy are relatively low. Importantly, based on the current results, we can only speculate on this aspect, given that we only find no support for age-related increases in mistrust and deception rather than direct support for developmental stability in each strategy.

Interestingly, even children playing with cooperative incentives mistrusted their coplayers in almost one-third of trials. This finding is surprising given that previous research among Western societies has yielded trustful attitudes under such conditions [5]. Again, one explanation lies in the familiarity of the coplayer. Children knew each other well, which may have shaped their mistrust even in the context of cooperative incentives. Children in Western societies are particularly mistrusting after experiencing the informant misleading them [26,5]. If participants knew that their coplayers deceived them previously and outside the study context, this may have led to an increased skepticism even in the context of cooperation.

As predicted, children's mistrust and deception were positively linked to one another on an individual level. Since none of the behaviors increased with age, this link does not lend clear support to claims on the co-development of selective trust and deception [4,8]. Both phenomena may be linked as a stable, trait-like disposition. The age range tested here is, however, limited. Further research is needed to assess incentive-based mistrust and deception outside Western, urbanized contexts to understand the phylogenetic and ontogenetic roots of these communicative strategies.

In sum, this study provides evidence that children's incentive-based mistrust is apparent among young children from a recent hunter-gatherer population. At the same time, children deceived their peers at low rates and regardless of whether incentives implied cooperation or competition. The ontogeny of social learning, including decisions about whom and when to mistrust or deceive, takes place in specific cultural contexts and between familiar individuals with diverse relationships. Understanding the role of these factors is crucial for understanding the ontogeny of human communication and social learning.

## Supporting information

**S1 Fig. Schematic procedure in both conditions.** E = Experimenter; S = Sender; R = Receiver; (a) S and R observe E putting the balls on the plates; (b) R turns around while E shows S the location of the reward (here: candy is on plate closer to E); (c) S places the stick on one plate to indicate the location of the reward (Deception assessed); (d) R chooses while S turns away (Mistrust assessed); (e) balls are stored according to R's choice before S turns back to the scene; hypothesis-conform behaviors are illustrated with honesty and trust during cooperation and deception and mistrust during competition.
(DOCX)

**S1 Table. Raw data.** (a) Hypothesis 1; (b) Hypothesis 2; (c) Hypothesis 3; Numbers in cells reflect number of children showing the respective behaviors; DEC = Sender deceives; HON = Senders gives honest hint; MIS = Receiver mistrusts the sender; TRU = Receiver trusts the sender; each child engaged in each role (sender & receiver) in two trials throughout the study and without receiving feedback between trials.
(DOCX)

**S2 Table. Model outputs for hypotheses 1 and 2,** $^*p < .05.$
(DOCX)

**S3 Table. Additional model to test whether correlation between mistrust and deception varies by age (Outcome/Vigilance: 1 = child deceives AND mistrusts; 0 = remaining children).**
(DOCX)

**S1 File. Codebook.**
(DOCX)

## Acknowledgments

We are grateful for all children who participated in our study, their parents, teachers, and the community at Ondera. We thank L. |Useb for help in data acquisition, translations, and coding; D. Tjizao for help with study translations; S. Peoples for helpful discussions on the procedure; T. Toppe and M. Thiele for helpful comments on previous versions of the manuscript; N. Blume for the visualization of the procedure; R. Raunijar for coding assistance

## Author Contributions

**Conceptualization:** Roman Stengelin.

**Data curation:** Roman Stengelin.

**Formal analysis:** Roman Stengelin, Robert Hepach.

**Funding acquisition:** Daniel B. M. Haun.

**Investigation:** Roman Stengelin.

**Methodology:** Roman Stengelin.

**Project administration:** Roman Stengelin.

**Supervision:** Robert Hepach, Daniel B. M. Haun.

**Visualization:** Roman Stengelin.

**Writing – original draft:** Roman Stengelin.

**Writing – review & editing:** Robert Hepach, Daniel B. M. Haun.

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
