## [Decision Letter · Decision Letter 0]

8 Nov 2019

PONE-D-19-28205

Hai||om children mistrust, but do not deceive, peers with opposing self-interests

PLOS ONE

Dear Mr. Stengelin,

Thank you for submitting your manuscript to PLOS ONE. After careful consideration, we feel that it has merit but does not fully meet PLOS ONE’s publication criteria as it currently stands. Therefore, we invite you to submit a revised version of the manuscript that addresses the points raised during the review process.

Two expert reviewers have closely read your manuscript, as have I. As you will see, the reviewers are enthusiastic about this paper, and both recommend minor revisions. I agree with them. The paradigm is clever, the results compelling, and the issue is clearly important and thus this paper makes a clear contribution to the literature. That said, each reviewer raised a number of important points that I would like you to consider. Both reviewers ask for quite a bit more detail in several areas, especially in the results and analysis sections and in the description of the population tested. Reviewer 2 also asks that you more clearly situate this paradigm within the relevant literature, discussing whether it or similar approaches have been used with other populations, and in particular how this might bear on your interpretation of the data. I do not have specific concerns to add to those the reviewers raise, however I do want you to make sure that you engage and respond thoroughly to each of the points they raise. I am considering this a minor revision given the recommendations of the reviewers, but I do not want you to take that as an indication that these concerns are minor and do not warrant serious consideration. They do, and I will expect that to be sufficiently addressed in your revised submission. 

We would appreciate receiving your revised manuscript by Dec 23 2019 11:59PM. To enhance the reproducibility of your results, we recommend that if applicable you deposit your laboratory protocols in protocols.io, where a protocol can be assigned its own identifier (DOI) such that it can be cited independently in the future. For instructions see: http://journals.plos.org/plosone/s/submission-guidelines#loc-laboratory-protocols

We look forward to receiving your revised manuscript.

Kind regards,

Lucas Payne Butler

Academic Editor

PLOS ONE

Journal Requirements:

1. During our internal checks, the in-house editorial staff noted that you conducted research or obtained samples in another country. Please check the relevant national regulations and laws applying to foreign researchers and state whether you obtained the required permits and approvals. Please address this in your ethics statement in both the manuscript and submission information.

2. Thank you for inlcuding your funding statement; "The funders had no role in study design, data collection and analysis, decision to publish, or preparation of the manuscript."

Please provide an amended Funding Statement that declares *all* the funding or sources of support received during this specific study (whether external or internal to your organization) as detailed online in our guide for authors at http://journals.plos.org/plosone/s/submit-now.  

Please state what role the funders took in the study.  If any authors received a salary from any of your funders, please state which authors and which funder. If the funders had no role, please state: "The funders had no role in study design, data collection and analysis, decision to publish, or preparation of the manuscript."

4. Please upload a copy of Figure 2, to which you refer in your text on page 10. If the figure is no longer to be included as part of the submission please remove all reference to it within the text.

5. We note you have included a table to which you do not refer in the text of your manuscript. Please ensure that you refer to Table  in your text; if accepted, production will need this reference to link the reader to the Table.

6. We note that Figure(s) [1] in your submission contain copyrighted images. All PLOS content is published under the Creative Commons Attribution License (CC BY 4.0), which means that the manuscript, images, and Supporting Information files will be freely available online, and any third party is permitted to access, download, copy, distribute, and use these materials in any way, even commercially, with proper attribution. For more information, see our copyright guidelines: http://journals.plos.org/plosone/s/licenses-and-copyright.

1.    You may seek permission from the original copyright holder of Figure(s) [#] to publish the content specifically under the CC BY 4.0 license.

Reviewers' comments:

Reviewer's Responses to Questions

**Comments to the Author**

1. Is the manuscript technically sound, and do the data support the conclusions?

Reviewer #1: Yes

Reviewer #2: Yes

2. Has the statistical analysis been performed appropriately and rigorously? 

Reviewer #1: Yes

Reviewer #2: Yes

3. Have the authors made all data underlying the findings in their manuscript fully available?

Reviewer #1: Yes

Reviewer #2: No

4. Is the manuscript presented in an intelligible fashion and written in standard English?

Reviewer #1: Yes

Reviewer #2: Yes

5. Review Comments to the Author

Reviewer #1: Summary: This ms presents a single study on non-Western (non WEIRD) participants' deception and mistrust. 64 children were tested on a novel paradigm in which they worked in dyads to either compete with or collaborate to obtain rewards. The DV was the # of times the clue-giving partner deceives and the # of times the clue-receiving partner mistrusted. Overall, the study finds no age or condition effects on deception, but that mistrust was impacted by competitive contexts. Moreover, deceitful players were also more likely to mistrust. This ms has many strengths, namely the clever paradigm, extension of cooperation/competition to new cultural groups, and the straightforwardly written ms and analyses. I have a few suggestions for revision:

1) Can the authors give a little more context about the cultural group studied here? They mention that they were hunter-gatherers until recently, but what does "recently" mean? Recently in historical vs. evolutionary time may mean very different things.

2) Authors mention that they predicted that in a culture in which there is a strong social norm to share in a compulsory manner, there may be less deception, but I would have predicted the opposite. In a culture in which there is a strong social norm to share, wouldn't there be a greater reason to lie about how much one has? I am either missing something, or this hypothesis is weakly specified. Can this be explained a little better in a revision?

3) How closely did children follow the experimenter's instructions? Were there any instances of cheating, or attempting to change the rules of the game? How did researchers deal with these instances if this were the case?

4) Analyses are straightforward, but a few additional details would, in my view, strengthen this ms:

a) can the authors compare deception rates to what has been found in prior literature in Western participants? It would be helpful to have a sense of what deception rates were like overall. Would also be helpful to give readers some #'s about deception and mistrust, and figures as well.

b) The correlation between mistrust and deceit is very interesting -- did this vary by age? I would guess that this type of self-other awareness would emerge with age (realizing that because one is likely to cheat, others are too). There might be a few ways to look at this with age - perhaps seeing if the participants who *both* mistrusted and deceived at least once, and looking at how proportions of this type of participant varies with age?

5) One question I had throughout this ms is what theory of mind or perspective taking looks like in this population. Granted, there may not be much data, but anything the authors can point to, would be helpful for giving some context to these data.

Reviewer #2: In this paper, the authors explore the ontogeny of mistrust and deception among 4 to 8 year old Hai||om children. There’s much to like about this paper: pre-registration, careful study design, open data, and a non-WEIRD participant pool. I have a few notes for the authors to consider and am recommending a revised resubmission.

1. I appreciate the difficulty of collecting these data and commend the authors on expanding out beyond a WEIRD participant pool. One question I have, however, is just how much of the observed differences are actually due to Hai||om cultural norms, as the authors argue. Has this paradigm been used in other cultural contexts, as the authors mentioned in their pre-registration? Are related paradigms from previous work close enough in content that they could be used to support the conclusions that there is “cross-cultural variability”? It would be useful for the authors to expand the introduction and discussion to better review in more methodological detail related studies that could support this point.

2. I appreciate that the authors have made their data openly available. However, there are a number of issues with the data file. First, I would advise that the format of the data file be updated. It is current at “.txt” file that is difficult to use in conjunction with statistical software. I’d like to request that a “.csv” file be added. Second, without a codebook, the data is hard to interpret. And third, the columns “Deception2” and “Mistrust2” are both blank and the data is missing. I assume these are the third & fourth trials, e.g. half of the study. And last, I would also ask that an annotated R script be added to the submission so that analyses can be reproduced.

3. Figure 2 is referenced in the text but is missing. I assume this is a visualization of the results? If not, I would request that the authors add in figures that can more readily communicate the results of the study to readers.

4. I would like to see more ethnographic background on the Hai||om communities the researchers worked with. They are described as being former hunter-gatherers, but I would like a small amount of review about their current subsistence patterns and cultural norms with more citations.

5. The authors mention that only children who passed comprehension checks were included. How many were excluded in total? Was this exclusion criterion pre-registered?

6. Did the practice trials result in actual rewards for the children or just simulated rewards? Given that the practice trials took place within the same dyads as the test trials, I wonder how behavior in these preceding tasks influences behavior in the test trials. Can the authors formally test this question using their data?

7. Were children age or gender-matched for their dyads? Was there any information collected on familiarity or relatedness between the children?

8. I would avoid the term “critical period” in line 56 as it has a specific definition in cognitive development that I do not think the authors are referring to.

9. There are a number of typos and grammatical errors throughout the manuscript; I'd recommend the manuscript be more closely proofread.

10. Instead of raw counts as in Table 1 (or at least, in addition to Table 1), I would like to see model outputs in either the main text or the Supplemental materials.

Dorsa Amir

Boston College

6. PLOS authors have the option to publish the peer review history of their article (what does this mean?). If published, this will include your full peer review and any attached files.

Reviewer #1: No

Reviewer #2: Yes: Dorsa Amir

---

## [Author Response · Author response to Decision Letter 0]

17 Dec 2019

Dear Professor Dr. Butler:

Thank you for the opportunity to revise our manuscript PONE-D-19-28205. We are grateful for the positive evaluation provided by yourself and the reviewers. 

“This ms has many strengths, namely the clever paradigm, extension of cooperation/competition to new cultural groups, and the straightforwardly written ms and analyses. “ 

(Reviewer 1)

“There’s much to like about this paper: pre-registration, careful study design, open data, and a non-WEIRD participant pool.” 

(Reviewer 2)

In the revised version of the manuscript we now address all the points raised by yourself and the reviewers. We outline the revisions we made point by point and have boldfaced the specific points.

You wrote:

(1)

“Both reviewers ask for quite a bit more detail in several areas, especially in the results and analysis sections and in the description of the population tested.”

We have added information on all these aspects (see below).

Descriptions of the study population:

“The Hai||om of northern Namibia are recent former hunter-gatherers and may thus be highly informative with regards to the ontogeny of mistrust and deception within peer interactions. Today, the Hai||om increasingly practice a mixed economy in which hunting and gathering is combined with agriculture, the selling of handmade crafts, and occasional wage labor (15). However, those living in more rural areas are still describing their cultural identity as that of hunter-gatherers, and their traditional values and socialization goals are still practiced and considered valuable. One foundational norm that the Hai||om and other hunter-gatherer societies typically emphasize is that of sharing on demand (15, see also 16,2). Accordingly, individuals are obliged to share resources among members of the community upon demand from an early age onwards (16). This norm may arguably shape the ontogeny of deception and mistrust such that more cooperative, rather than competitive, communication strategies may be promoted when facing limited resources. That is, if obtained resources are shared among peers regardless of individual merit or success (16), incentives for deception (and, as a consequence, mistrust) should be lowered markedly as compared to societies emphasizing merit and individual attainments. Accordingly, such egalitarian sharing norms may create cooperative incentives under conditions that would be perceived as competitive among Western populations.“ (p. 4; see also reviewer 2, comment 4)

“All participants were Hai||om children from Ondera, a rural village with around 800 inhabitants in northern Namibia. While some people at Ondera find employment at the local vegetable farm, others sell handmade crafts at Etosha Park and cultivate plants in their gardens. Gathering bushfood is practiced commonly by the Hai||om at Ondera. Hunting is forbidden by law. In line with Hai||om traditional parenting practices, children are given high levels of autonomy from early on (15). As is common in hunter-gatherer societies, children’s daily activities are typically structured within mixed-age peer groups, which is why peer interactions can be considered essential for social learning (see also 18).” 

(p. 6; see also reviewer 2, comment 4)

Descriptions of analyses and results:

 “

Fig 2: Children’s behaviors throughout the study; (a) honesty vs. deception as senders; (b) trust vs. mistrust as receivers” (p. 12; see also reviewer 2, comment 3)

“S3: Model Outputs

Fixed Effects H1: Deception as Sender H2: Mistrust as Receiver

 Estimate SE p Estimate SE p

Predictors 

(Intercept) 0.848 1.602 .597 -0.552 1.472 .708

Age -0.286 0.283 .231 0.148 0.216 .493

Condition -0.482 0.473 .309 -1.077 0.424 .011*

Controls 

Sex -0.342 0.487 .483 -0.152 0.422 .719

Trial 0.104 0.456 .820 -0.510 0.388 .189

S2: Model outputs for hypotheses 1 and 2, *p < .05” 

(p. 2, supplementary materials I; see also reviewer 2, comment 10)

“S4: 

Fixed Effects Vigilance in either role

 Estimate SE p

Predictors 

(Intercept) -0.003 1.929 .999

Age -0.124 0.277 .654

S4: Additional model to test whether correlation between mistrust and deception varies by age (Outcome/Vigilance: 1 = child deceives AND mistrusts; 0 = remaining children)“ 

(p. 2, supplementary material I; see also reviewer 1, comment 4b)

(2)

“Reviewer 2 also asks that you more clearly situate this paradigm within the relevant literature, discussing whether it or similar approaches have been used with other populations, and in particular how this might bear on your interpretation of the data.”

We now situate the paradigm more clearly within the relevant literature on the ontogeny of selective trust and deception in young children from western populations and reflect on this in the discussion:

“These findings differ from previous studies assessing the early ontogeny of mistrust and deception based on procedures incentivizing competition. In previous studies, preschool children from urban Western populations were found to both mistrust and deceive their coplayers at similarly high rates (3,4). While children in the current study mistrusted their competitive coplayers at rates comparable to previous studies, they used deception less frequently and irrespective of whether game incentives favored competition or cooperation. Moreover, neither mistrust nor deception became more frequent with age, which also stands in contrast to previous work (3–5). Procedural details may account for these diverging results: While previous studies introduced either unfamiliar adults (4) or puppets played by adults (3,5) as coplayers, children in the current study interacted with a familiar peer. It is plausible that senders provided receivers with honest cues when playing with a competitive incentive because of reputational concerns (see below). […]“ (p. 12)

Reviewer 1

(1)

“Can the authors give a little more context about the cultural group studied here? They mention that they were hunter-gatherers until recently, but what does "recently" mean? Recently in historical vs. evolutionary time may mean very different things.“

In the revised form of the manuscript we now provide more ethnographic information on the community in which we conducted the study, including some information on their subsistence:

“The Hai||om of northern Namibia are recent former hunter-gatherers and may thus be highly informative with regards to the ontogeny of mistrust and deception within peer interactions. Today, the Hai||om increasingly practice a mixed economy in which hunting and gathering is combined with agriculture, the selling of handmade crafts, and occasional wage labor (15). However, those living in more rural areas are still describing their cultural identity as that of hunter-gatherers, and their traditional values and socialization goals are still practiced and considered valuable. One foundational norm that the Hai||om and other hunter-gatherer societies typically emphasize is that of sharing on demand (15, see also 16,2). Accordingly, individuals are obliged to share resources among members of the community upon demand from an early age onwards (16). This norm may arguably shape the ontogeny of deception and mistrust such that more cooperative, rather than competitive, communication strategies may be promoted when facing limited resources. That is, if obtained resources are shared among peers regardless of individual merit or success (16), incentives for deception (and, as a consequence, mistrust) should be lowered markedly as compared to societies emphasizing merit and individual attainments. Accordingly, such egalitarian sharing norms may create cooperative incentives under conditions that would be perceived as competitive among Western populations.“ (p. 4)

“All participants were Hai||om children from Ondera, a rural village with around 800 inhabitants in northern Namibia. While some people at Ondera find employment at the local vegetable farm, others sell handmade crafts at Etosha Park and cultivate plants in their gardens. Gathering bushfood is practiced commonly by the Hai||om at Ondera. Hunting is forbidden by law. In line with Hai||om traditional parenting practices, children are given high levels of autonomy from early on (15). As is common in hunter-gatherer societies, children’s daily activities are typically structured within mixed-age peer groups, which is why peer interactions can be considered essential for social learning (see also 18).” (p. 6)

(2)

“Authors mention that they predicted that in a culture in which there is a strong social norm to share in a compulsory manner, there may be less deception, but I would have predicted the opposite. In a culture in which there is a strong social norm to share, wouldn't there be a greater reason to lie about how much one has? I am either missing something, or this hypothesis is weakly specified. Can this be explained a little better in a revision?

Given that children did not earn their rewards in private, but in the presence of their coplayer, deceiving the opponent by hiding one’s earnings was not possible in the current study. We now clarify this argument and discuss it more thoroughly:

“The societal emphasis on sharing on demand among the Hai||om may also be essential to understand why senders in this study did not deceive receivers at higher rates. Accordingly, children may have hinted at the correct location of the reward because they did not adopt a competitive stance toward their coplayer in the game. Children may have expected that resources obtained throughout the game would be shared afterward regardless of whether they or their coplayer would obtain the reward. This expectation may have led them to give honest hints to avoid frictions. However, this conclusion fails to explain children’s mistrust in the role of receivers. If children had a cooperative stance in general and regardless of condition, one would expect them to trust their coplayer’s hints in either condition.” (p. 13)

(3)

“How closely did children follow the experimenter's instructions? Were there any instances of cheating, or attempting to change the rules of the game? How did researchers deal with these instances if this were the case?“

We did not observe any attempts of modifying the game rules throughout the study. An adult experimenter was present throughout the study, which may have led the participants to adhere to the game instructions. The only notable behavior that occurred in this regard was that a few children attempted to peak during the initial training session in which they made their decisions strictly randomly. If so, the experimenter reminded children to refrain from doing so. During the test phase, we did not observe any peaking behaviors. To avoid misinterpretations of the current procedure, we now give this information in the manuscript. We briefly mention this procedural detail in the manuscript:

“If children attempted to peak to see their partner’s choice during this phase, they were reminded by E to turn away to ensure children’s privacy in making their decisions.“ (p. 8)

(4a)

“Can the authors compare deception rates to what has been found in prior literature in Western participants? It would be helpful to have a sense of what deception rates were like overall. Would also be helpful to give readers some #'s about deception and mistrust, and figures as well“

We have added a detailed discussion on how the current results can be related to previous research:

“These findings differ from previous studies assessing the early ontogeny of mistrust and deception based on procedures incentivizing competition. In previous studies, preschool children from urban Western populations were found to both mistrust and deceive their coplayers at similarly high rates (3,4). While children in the current study mistrusted their competitive coplayers at rates comparable to previous studies, they used deception less frequently and irrespective of whether game incentives favored competition or cooperation. Moreover, neither mistrust nor deception became more frequent with age, which also stands in contrast to previous work (3–5). Procedural details may account for these diverging results: While previous studies introduced either unfamiliar adults (4) or puppets played by adults (3,5) as coplayers, children in the current study interacted with a familiar peer. It is plausible that senders provided receivers with honest cues when playing with a competitive incentive because of reputational concerns (see below). […]“ (p. 12)

(4b)

“The correlation between mistrust and deceit is very interesting -- did this vary by age? I would guess that this type of self-other awareness would emerge with age (realizing that because one is likely to cheat, others are too). There might be a few ways to look at this with age - perhaps seeing if the participants who *both* mistrusted and deceived at least once, and looking at how proportions of this type of participant varies with age? “

We have added an additional analysis to the supplementary materials following the reviewer’s suggestion. Running a generalized linear model with age as a predictor and children’s vigilance (1 if children deceive as senders and mistrust as receivers; 0 for all remaining cases) does not reveal a significant effect of age (χ2 (1) = 0.20, p = .654).

(5)

“One question I had throughout this ms is what theory of mind or perspective taking looks like in this population. Granted, there may not be much data, but anything the authors can point to, would be helpful for giving some context to these data.“

There is, to our knowledge, no empirical data on the ontogeny of Theory of Mind among Hai||om children. The only study that we are aware of that assessed Theory of Mind (in particular: False belief-reasoning) among hunter-gatherer children is that of Avis & Harris (1991). We agree that Theory of Mind/false belief is a central prerequisite for incentive-based mistrust and deception and have included a paragraph in which we highlight this more explicitly.

“Alternatively, children may have provided truthful information to their coplayers because they lacked the social-cognitive skills needed for deception, such as false belief-reasoning (4,21). However, children’s selective mistrust as receivers suggests that they understood that the game’s incentives might lead their coplayer to deceive them. As such, the low rates of deception in the current study are unlikely due to a lack of skills in false belief-reasoning. To our knowledge, there is no documentation of how false belief-reasoning emerges develops among Hai||om children. The one study that has assessed the ontogeny of this skill among hunter-gatherer children has documented sophisticated skills in false belief-reasoning among Aka children (22), which is in line with other evidence suggesting that such reasoning is common among young children from rural, non-Western societies (23,24,but see 25). Future studies will need to assess if the links between false belief-reasoning and deception (and mistrust) observed among Western children (21) recur among non-Western societies. Based on the current findings, it is unlikely that a lack of skills in false belief-reasoning is the reason for the low rates of deception in the competition condition.”

(p. 13)

 

Reviewer #2: 

(1)

“I appreciate the difficulty of collecting these data and commend the authors on expanding out beyond a WEIRD participant pool. One question I have, however, is just how much of the observed differences are actually due to Hai||om cultural norms, as the authors argue. Has this paradigm been used in other cultural contexts, as the authors mentioned in their pre-registration? Are related paradigms from previous work close enough in content that they could be used to support the conclusions that there is “cross-cultural variability”? It would be useful for the authors to expand the introduction and discussion to better review in more methodological detail related studies that could support this point.“

To our knowledge, previous studies have not yet observed children’s mistrust and deception toward peer coplayers. Typically, these studies assessed children’s mistrust and deception in paradigms in which children constantly received feedback on whether their behavior resulted in access to rewards and in which children were explicitly instructed on the possibility to deceive and/or mistrust (e.g., Reyes-Jaquez & Echols, 2015; Sher, Koenig, & Rustichini, 2014). Moreover, these studies assessed children’s mistrust and deception toward adult strangers (or puppets), a scenario that would lack ecological validity in the cultural context assessed here. Although predicting exact rates of deception in the current paradigms is thus difficult, evidence taken from these studies suggests that the majority of children are capable of using strategic deception at around their fifth year of live (e.g., Reyes-Jaquez & Echols, 2015; Sher, Koenig, & Rustichini, 2014). 

We initially referred to “cross-cultural variability” with regard to the ontogenetic stability in mistrust and deception as observed in this study, rather than in the absolute levels of deception in children. We have refined these conclusions and have added information to embed the current paradigm into the scientific literature:

“These findings differ from previous studies assessing the early ontogeny of mistrust and deception based on procedures incentivizing competition. In previous studies, preschool children from urban Western populations were found to both mistrust and deceive their coplayers at similarly high rates (3,4). While children in the current study mistrusted their competitive coplayers at rates comparable to previous studies, they used deception less frequently and irrespective of whether game incentives favored competition or cooperation. Moreover, neither mistrust nor deception became more frequent with age, which also stands in contrast to previous work (3–5). Procedural details may account for these diverging results: While previous studies introduced either unfamiliar adults (4) or puppets played by adults (3,5) as coplayers, children in the current study interacted with a familiar peer. It is plausible that senders provided receivers with honest cues when playing with a competitive incentive because of reputational concerns (see below). […]“ (p. 12)

 (2)

“I appreciate that the authors have made their data openly available. However, there are a number of issues with the data file. First, I would advise that the format of the data file be updated. It is current at “.txt” file that is difficult to use in conjunction with statistical software. I’d like to request that a “.csv” file be added. Second, without a codebook, the data is hard to interpret. And third, the columns “Deception2” and “Mistrust2” are both blank and the data is missing. I assume these are the third & fourth trials, e.g. half of the study. And last, I would also ask that an annotated R script be added to the submission so that analyses can be reproduced.“

We have added the complete data as a .csv-document and modified our code accordingly. As a supplementary material, we have also added a codebook to allow for an easier understanding of this datafile. The codes used for running the analyses are also made available within the current submission.

(3)

“Figure 2 is referenced in the text but is missing. I assume this is a visualization of the results? If not, I would request that the authors add in figures that can more readily communicate the results of the study to readers. “

We thank the reviewer for this note. Indeed, the reference to “Figure 2” was misleading as we aimed at referring to “Table 1” instead, in which we visualize the raw data within the manuscript. In line with some other comments of the reviewers, we have decided to include a Figure in which we plot children’s behavior in either condition and have moved Table 1 into the supplementary material.

“

Fig 2: Children’s behaviors throughout the study; (a) honesty vs. deception as senders; (b) trust vs. mistrust as receivers” (p. 12)

(4)

“I would like to see more ethnographic background on the Hai||om communities the researchers worked with. They are described as being former hunter-gatherers, but I would like a small amount of review about their current subsistence patterns and cultural norms with more citations.“

We have added more ethnographic information on the community in the revised manuscript. There are currently no specific ethnographic descriptions of the Hai||om at Ondera, given that the settlement was only established a few years ago. We thus use descriptions from other Hai||om communities and other hunter-gatherer populations sharing similar emphasizes as the Hai||om at Ondera in our manuscript and combine it with our own observations and residents’ descriptions.

“The Hai||om of northern Namibia are recent former hunter-gatherers and may thus be highly informative with regards to the ontogeny of mistrust and deception within peer interactions. Today, the Hai||om increasingly practice a mixed economy in which hunting and gathering is combined with agriculture, the selling of handmade crafts, and occasional wage labor (15). However, those living in more rural areas are still describing their cultural identity as that of hunter-gatherers, and their traditional values and socialization goals are still practiced and considered valuable. One foundational norm that the Hai||om and other hunter-gatherer societies typically emphasize is that of sharing on demand (15, see also 16,2). Accordingly, individuals are obliged to share resources among members of the community upon demand from an early age onwards (16). This norm may arguably shape the ontogeny of deception and mistrust such that more cooperative, rather than competitive, communication strategies may be promoted when facing limited resources. That is, if obtained resources are shared among peers regardless of individual merit or success (16), incentives for deception (and, as a consequence, mistrust) should be lowered markedly as compared to societies emphasizing merit and individual attainments. Accordingly, such egalitarian sharing norms may create cooperative incentives under conditions that would be perceived as competitive among Western populations. “ (p. 4)

“All participants were Hai||om children from Ondera, a rural village with around 800 inhabitants in northern Namibia. While some people at Ondera find employment at the local vegetable farm, others sell handmade crafts at Etosha Park and cultivate plants in their gardens. Gathering bushfood is practiced commonly by the Hai||om at Ondera. Hunting is forbidden by law. In line with Hai||om traditional parenting practices, children are given high levels of autonomy from early on (15). As is common in hunter-gatherer societies, children’s daily activities are typically structured within mixed-age peer groups, which is why peer interactions can be considered essential for social learning (see also 18).” (p. 6)

(5)

“The authors mention that only children who passed comprehension checks were included. How many were excluded in total? Was this exclusion criterion pre-registered?“

We did not refer to the comprehension questions in the preregistration but included this criterion in order to ensure that participants would understand the game’s incentives, without which the assessment of mistrust and deception would not reveal meaningful insights. 

Two dyads (one female and one male) were excluded from the study because one child did not solve the comprehension checks before Guessing Game I. In both cases, we did not continue with the procedure hereafter.

“If one of the participants gave an incorrect answer, E explained the rules of the game once more, before repeating the comprehension questions. Only dyads in which both children answered the comprehension questions correctly were included in the study. Two dyads did not solve these questions and were thus not examined any further.” (p. 8)

(6)

“Did the practice trials result in actual rewards for the children or just simulated rewards? Given that the practice trials took place within the same dyads as the test trials, I wonder how behavior in these preceding tasks influences behavior in the test trials. Can the authors formally test this question using their data?“

Indeed, children earned rewards during practice trials. During Guessing Game II, these rewards were manipulated such that each child received candy for two out of four trials. Accordingly, these rewards cannot have impacted children’s subsequent behaviors differently between conditions. We did not manipulate the rewards children received during Guessing Game I because of the immediate feedback children received after each trial during this phase. We coded the rewards children won during Guessing Game I and assessed whether their mistrust and deception were affected by this variable. Neither children’s mistrust (χ2 (1) = 0.371, p = .543) nor their attempts to deceive (χ2 (1) = 0.348, p = .555) varied based on the rewards they received during Guessing Game I.

(7)

“Were children age or gender-matched for their dyads? Was there any information collected on familiarity or relatedness between the children?“

Children were matched for sex and age throughout the study. We did not obtain any information on familiarity between children, but due to the small community size at Ondera all children knew each other well. We did not test dyads containing of siblings or children growing up in the same household in the current study, but cannot rule out that some children were otherwise related. 

“Children within dyads were matched for age (MDiff (SD)= 0.47 years (0.44)) and sex. Siblings or children living in one household were not tested together in the same dyad.” (p. 6)

(8)

“I would avoid the term “critical period” in line 56 as it has a specific definition in cognitive development that I do not think the authors are referring to.“

We have modified this sentence accordingly:

“This development of children’s selective mistrust is accompanied by an increase in their use of strategic deception.” (p. 3)

(9)

“There are a number of typos and grammatical errors throughout the manuscript; I'd recommend the manuscript be more closely proofread.“

We have proofread the current version of the manuscript to correct typos and grammatical errors.

(10)

“Instead of raw counts as in Table 1 (or at least, in addition to Table 1), I would like to see model outputs in either the main text or the Supplemental materials.“

We have added the model outputs to the supplementary materials:

“S3: Model Outputs

Fixed Effects H1: Deception as Sender H2: Mistrust as Receiver

 Estimate SE p Estimate SE p

Predictors 

(Intercept) 0.848 1.602 .597 -0.552 1.472 .708

Age -0.286 0.283 .231 0.148 0.216 .493

Condition -0.482 0.473 .309 -1.077 0.424 .011*

Controls 

Sex -0.342 0.487 .483 -0.152 0.422 .719

Trial 0.104 0.456 .820 -0.510 0.388 .189

S2: Model outputs for hypotheses 1 and 2, *p < .05” (p. 2, supplementary materials I)

---

## [Decision Letter · Decision Letter 1]

28 Jan 2020

PONE-D-19-28205R1

Hai||om children mistrust, but do not deceive, peers with opposing self-interests

PLOS ONE

Dear Mr. Stengelin,

Thank you for submitting your manuscript to PLOS ONE. After careful consideration, we feel that it has merit but does not fully meet PLOS ONE’s publication criteria as it currently stands. Therefore, we invite you to submit a revised version of the manuscript that addresses the points raised during the review process.

Thank you for your detailed attention to the concerns raised in the previous round of review. As you'll see, both reviewers believe you substantively addressed their concerns, and both recommend publication. I agree with them. At this point, there is really only one minor comment from the second reviewer that I would ask you to address: "One minor comment: I would ask that the authors append the analyses they performed in response to my comments (e.g. the practice trial analysis) into the actual manuscript or supplement itself so that it's accessible to readers." 

We would appreciate receiving your revised manuscript by Mar 13 2020 11:59PM. To enhance the reproducibility of your results, we recommend that if applicable you deposit your laboratory protocols in protocols.io, where a protocol can be assigned its own identifier (DOI) such that it can be cited independently in the future. For instructions see: http://journals.plos.org/plosone/s/submission-guidelines#loc-laboratory-protocols

We look forward to receiving your revised manuscript.

Kind regards,

Lucas Payne Butler

Academic Editor

PLOS ONE

Reviewers' comments:

Reviewer's Responses to Questions

**Comments to the Author**

1. If the authors have adequately addressed your comments raised in a previous round of review and you feel that this manuscript is now acceptable for publication, you may indicate that here to bypass the “Comments to the Author” section, enter your conflict of interest statement in the “Confidential to Editor” section, and submit your "Accept" recommendation.

Reviewer #1: All comments have been addressed

Reviewer #2: All comments have been addressed

2. Is the manuscript technically sound, and do the data support the conclusions?

Reviewer #1: Yes

Reviewer #2: Yes

3. Has the statistical analysis been performed appropriately and rigorously? 

Reviewer #1: Yes

Reviewer #2: Yes

4. Have the authors made all data underlying the findings in their manuscript fully available?

Reviewer #1: Yes

Reviewer #2: Yes

5. Is the manuscript presented in an intelligible fashion and written in standard English?

Reviewer #1: Yes

Reviewer #2: Yes

6. Review Comments to the Author

Reviewer #1: I believe the authors have addressed previous comments. Generally, we need more work done with non-Western samples and this study will add nicely to the literature.

Reviewer #2: I appreciate the authors engaging thoughtfully with my feedback and am generally satisfied with their edits. One minor comment: I would ask that the authors append the analyses they performed in response to my comments (e.g. the practice trial analysis) into the actual manuscript or supplement itself so that it's accessible to readers. Pending this, I'm happy to recommend an acceptance of the manuscript.

7. PLOS authors have the option to publish the peer review history of their article (what does this mean?). If published, this will include your full peer review and any attached files.

Reviewer #1: Yes: Nadia Chernyak

Reviewer #2: Yes: Dorsa Amir

---

## [Author Response · Author response to Decision Letter 1]

3 Feb 2020

Reviewer 2:

“One minor comment: I would ask that the authors append the analyses they performed in response to my comments (e.g. the practice trial analysis) into the actual manuscript or supplement itself so that it's accessible to readers.“

We now mention this analysis in the manuscript. We have added random intercepts of ID to these models to account for within-subject variance. The results remain unaffected by doing so.

“The number of rewards children received during Guessing Game I was not included to the model because of convergence issues. In a separate analysis, this variable was not linked to children’s deception (χ2 (1) = 0.354, p = .552).“ (p. 11)

“The number of rewards children received during Guessing Game I did not predict their subsequent mistrust (χ2 (1) = 0.327, p = .568).“ (p. 11)

---

## [Editor Report · Decision Letter 2]

21 Feb 2020

Hai||om children mistrust, but do not deceive, peers with opposing self-interests

PONE-D-19-28205R2

Dear Dr. Stengelin,

We are pleased to inform you that your manuscript has been judged scientifically suitable for publication and will be formally accepted for publication once it complies with all outstanding technical requirements.

With kind regards,

Lucas Payne Butler

Academic Editor

PLOS ONE
---

## [Editor Report · Acceptance letter]

27 Feb 2020

PONE-D-19-28205R2 

Hai||om children mistrust, but do not deceive, peers with opposing self-interests 

Dear Dr. Stengelin:

I am pleased to inform you that your manuscript has been deemed suitable for publication in PLOS ONE. Congratulations! Your manuscript is now with our production department. 

With kind regards,

on behalf of

Dr. Lucas Payne Butler 

Academic Editor

PLOS ONE